# A highly efficient segmentation method for abdominal multi-organs on laptop

Junchen Xiong[1][0009−0000−1988−1184], Pengju Lyu[1][0009−0004−0863−8110], Tingyi Lin[1][0009−0006−1677−9524], Kehan Song[1][0009−−0002−9054−4871], Cheng Wang[1][0000−0003−1138−337X], and Jianjun Zhu[1,2†][0000−0001−5895−7663]

[1] Hanglok-Tech Co., Ltd., Hengqin 519000, China
[2] Zhongda Hospital, Medical School, Southeast University, Nanjing 210009, China
{jj.zhu}@hanglok-tech.cn

**Abstract.** A precise and real-time abdominal multi-organ segmentation method is of crucial importance for its practical application. In this study, we use a two-phase strategy to address this issue. In the phase one, we quickly localize the abdominal region, while the second phase focuses on fine segmentation of this region. This work builds upon last year's efforts. To improve inference efficiency, we designed a Lightweight Attention-based Convolutional Block for the phase two and incorporated it into the decoder. Additionally, the preprocessing process has been further optimized. The results on leaderboard validated promising performance, achieving an average score of 90.02% and 95.51% for the DSC and NSD. Additionally, the method's average running time on public validation is 16.34s in our laptop. In summary, this strategy effectively ensures the possibility of achieving high precision with low latency. Our code is available at: https://github.com/JCXiong1227/FLARE2024.

**Keywords:** Two-phase · Inference efficiency · Lightweight · Preprocessing.

## 1 Introduction

In the field of medical analysis, 3D CT-based multi-organ segmentation of the abdomen is of great clinical significance for disease treatment. In the past challenges organized by MICCAI FLARE[21], many methods [30] [16] have achieved satisfactory performance in both inference speed and accuracy using single Graphics Processing Unit (GPU). However, on laptops or hospital imaging edge devices where GPU resources may not be available, their temporal efficiency may be limited. Currently, there is rarely consideration of applying the their methods on CPU-based devices in the domain of abdominal multi-organ segmentation. Thus, achieving low-latency inference speeds on laptops is both a pioneering and highly significant research challenge.

In the early stages of deep learning, convolutional neural network frameworks, exemplified by U-Net, held milestone significance in the research of medical image

---

† Corresponding authors.

segmentation. U-Net[26] effectively compensates for the loss of fine-grained information during downsampling process of encoder by utilizing skip connections in the decoding stage. Subsequently, some methods [12] [24] have built upon this concept by enhancing feature representation through multi-scale within blocks or multi-path information aggregation, thereby making the model more robust and capable of handling increasingly intricate patterns and variations in medical images. Furthermore, some researchers believe that the long-range dependency limitations of CNNs may lead to suboptimal solutions. Therefore, they propose integrating Transformers with U-Net to capture both global and local information [9] [28] [36]. Their approaches allow the model to comprehend the global anatomical structure and local details of the medical image. These methods focus excessively on improving accuracy while neglecting inference time, making it difficult to apply them to common hardware devices. Xie et al. [32] and Gao et al. [6] leverage the advantages of CNNs and Transformers to achieve the balance between segmentation accuracy and efficiency. With the development of lightweight models, depthwise separable convolutions and model pruning have been applied to 3D model design. Excessive use of lightweight techniques may lead to unintended accuracy loss. Chen et al. [2] introduced dilated convolutions in module design. Zhao et al. [35] employed a teacher-student architecture, using 3D nnU-Net to distill a lightweight model. Liao et al. [15] utilized Mamba to achieve long-range spatial dependencies with linear complexity, in contrast to the transformer architecture. These methods have accelerated inference speed, but for abdominal multi-organ segmentation, there is still potential to further enhance real-time performance without compromising accuracy. Wu et al. [30] and Lyu et al. [16] adopted a two-stage training framework, where phase one locates the foreground, and phase two focuses solely on predicting this region. This approach filters out a significant amount of background area, offering a novel strategy to accelerate abdominal multi-organ segmentation.

In this work, we aim to develop a fast, low-resource, and accurate organ segmentation framework on laptop. To achieve this goal, we employed a two-stage network. In the phase one, we designed our block including in encoder using partial convolution and lightweight SegFormer head collaboratively achieve foreground localization quickly. In the phase two, a novel CNN-Transformer model were proposed. It adopts a scale-aware modulator and self-attention within the encoder blocks. To accelerate inference progress, we utilized asymmetric convolutions and group convolutions in the decoder. The results on the validation submission indicate that we superior performance while maintaining a fast inference speed on laptop, effectively balancing both aspects.

## 2   Method

The framework is implemented as a cascade of two networks (see Fig. 1), as demonstrated by previous works [30] [16], which have proven its efficacy in accelerating model inference. The data flow during inference is as follows: Phase one quickly segments the foreground region and uses it as input of phase two.

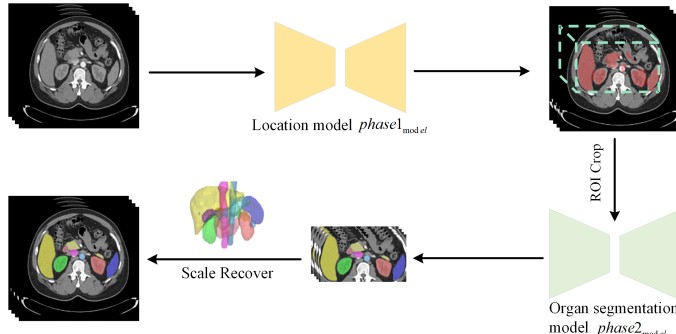

**Fig. 1.** An overview of the two-phase cascade network.

This approach filters out the regions unrelated to the abdomen, thereby reducing the time required for segmentation compared to using the network that involves only phase two.

### 2.1   Preprocessing

Considering that this challenge is executed on a laptop, we meticulously ordered the data preprocessing steps to minimize time delays. In the first phase, we resize the image dimensions to (128, 128, 128) before performing Z-normalization. If the order were reversed, the Z-normalization process on the entire input image would take significantly longer compared to doing so on the resized image. Following this guideline, in the phase two, we first extract the foreground region of the image based on the label, then uniformly resample the spatial spacing to (1.5mm, 1.5mm, 2mm), and perform Z-normalization. To enhance the model's robustness, we apply random flipping, random rotation, random affine transformations, random intensity shifting (offset: 0.1), and random scaling (scaling factor: 0.1) during the training stage. Subsequently, for each transformed image, we randomly crop six cubes with a 5:1 ratio of positive to negative samples, each of size (96, 96, 96), and input them into the model.

### 2.2   Proposed Method

**Network** This work is a further continuation of the research conducted by last year[16], aiming to accelerate the model's inference speed using lightweight techniques. Detailed information can be found in Fig. 2. $Phase1_{model}$ and $Phase2_{model}$ share common encoder including four stages. Firstly, the base channel numbers in the stem block are set to 32/60 for $Phase1_{model}$ and $Phase2_{model}$, respectively. Next, the number of channels progressively doubles, and the feature map size is halved compared to the preceding stage.

For phase one, we tested last year's work[16] and found that it can quickly locate the foreground region within one second on laptop. The primary reason is

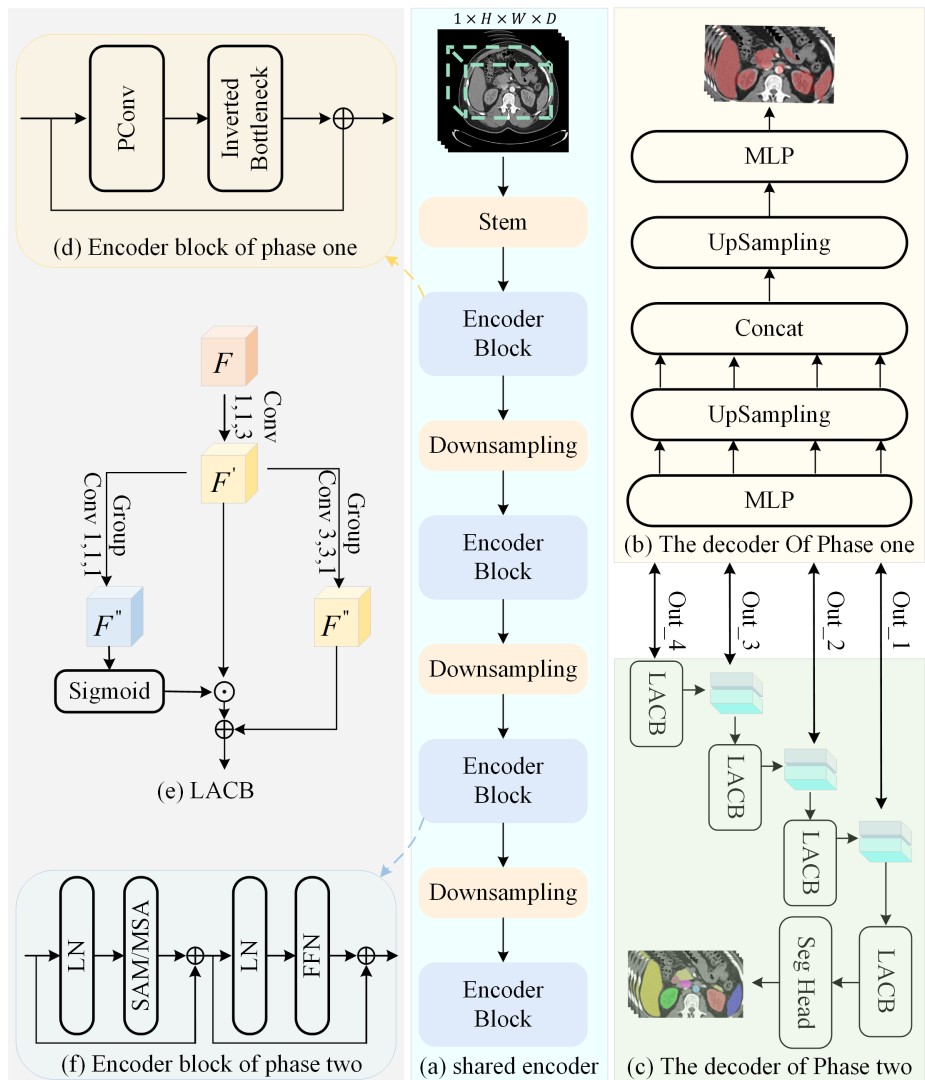

**Fig. 2.** The cascade method of proposed models. (a) The shared encoder backbone. (b) $Phase1_{model}$ decoder from [31]. (c) Lighted decoder for $Phase2_{model}$. (d) Encoder block in $Phase1_{model}$. (e) Lighted block in the decoder of $Phase2_{model}$. (f) Encoder block in $Phase2_{model}$.

that $Phase1_{model}$ employs partial convolutions and inverted bottlenecks during the encoding phase, which have been explicitly proven to accelerate inference speed in the works [3] [25]. Additionally, the decoder design for phase one is also quite streamlined, employing only the MLP decoder[31]. In summary, this design

has perfectly met the requirements for phase 1 with low latency, and therefore, no further modifications are needed.

For phase two, considering the influence of multi-scale local information and global context dependencies on accuracy in encoder, the Scale-Aware Modulator (SAM) and Multi-head Self-Attention (MSA) were designed[16]. The former leverages multi-scale approaches to extract local detail information in shallower layers, while the latter focuses more on global semantic information in deeper layers. To avoid affecting the model's real-time performance, this work employs depth-wise convolutions for the various kernels in the SAM module. The only distinction lies in the decoder design, where asymmetric convolutions and group convolutions are utilized to create a Lightweight Attention-based Convolutional Block (LACB), as shown in Fig. 2 (e). In LACB, the 1×1×3 and 3×3×1 convolution kernels were used individually to process inter-slice and intra-slice information, respectively. To reduce Flops computations, we employed group convolutions for processing inter-slice information. This operation may adversely affect accuracy, so we further incorporated a spatial attention mechanism within the LACB.

**Loss function** The choice appropriately of loss function is a crucial component in deep learning. In our experiments, we utilized two classical loss functions: the Dice loss function (Equation 1) and the cross-entropy loss function (Equation 2), which were combined with a 1:1 weight ratio (Equation 3) to train the model.

$$L_{ce} = -\frac{1}{C} \sum_{c=1}^{C} Y \log (P) \tag{1}$$

$$L_{dice} = \frac{1}{C} \sum_{c=1}^{C} \frac{2 \times (Y \cap P)}{Y \cup P} \tag{2}$$

$$L_{total} = L_{ce} + L_{dice} \tag{3}$$

Where, $C$ denotes totoal number of classes, and $Y$ is the one-hot encoding of ground-truth with $C$ classes.

### 2.3 Post-processing

Due to the differences in preprocessing applied in the networks ($phase1_{model}$ and $phase2_{model}$), the post-processing procedures also vary. In phase one, the segmentation results (128×128×128) must be rescaled to the original input dimensions. Subsequently, erroneous voxel regions (those smaller than 20×20×20) are filtered out to ensure the correct acquisition of foreground areas. For phase two, considering real-time performance, we first preserve solely the largest components of organs for the prediction results inferring by the cropped image regions, then restore the voxel spacing, and finally obtain the segmentation results at the original image size.

## 3    Experiments

### 3.1    Dataset and evaluation measures

The dataset is curated from more than 40 medical centers under the license permission, including TCIA [4], LiTS [1], MSD [27], KiTS [10, 11], autoPET [8, 7], AMOS [14], AbdomenCT-1K [23], TotalSegmentator [29], and past FLARE challenges [19, 20, 22]. The training set includes 2050 abdomen CT scans where 50 CT scans with complete labels and 2000 CT scans without labels. The validation and testing sets include 250 and 300 CT scans, respectively. The annotation process used ITK-SNAP [34], nnU-Net [13], MedSAM [17], and Slicer Plugins [5, 18].

The evaluation metrics encompass two accuracy measures—Dice Similarity Coefficient (DSC) and Normalized Surface Dice (NSD)—alongside one efficiency measures—runtime. These metrics collectively contribute to the ranking computation. During inference, GPU is not available where the algorithm can only rely on CPU.

### 3.2    Implementation details

**Environment settings** Throughout the entire experimental process, The hardware facilities and code execution-related tools or libraries we utilized are presented in Table 1.

**Table 1.** Development environments and requirements.

| System | Ubuntu 20.04.5 LTS |
|---|---|
| CPU | Intel (R) Xeon (R) Platinum 8358 CPU @ 2.60GHz |
| RAM | 1.0 Ti; 3200 MT/S |
| GPU (number and type) | NVIDIA A800 80G |
| CUDA version | 11.8 |
| Programming language | Python 3.8.15 |
| Deep learning framework | torch 2.0.1, torchvision 0.15.2 |
| Specific dependencies | monai 1.3.2 |
| Code | https://github.com/***** |

**Training protocols** Before initiating model training, we configured the necessary hyperparameters and the optimizer for the training protocols. The Adam optimizer, with a weight decay of $1e^{-5}$, was utilized across both training phases. The initial learning rate was set to $1e^{-3}$, and a cosine annealing strategy was employed for adjusting the learning rate. Each phase was trained for 150 epochs with a batch size of 6, and was supervised using the Dice coefficient and cross-entropy loss functions. There are also differences in the training settings between

the two phases. Specifically, $Phase1_{model}$ involves performing coarse segmentation on resized images, whereas $Phase2_{model}$ utilizes a fixed-size patch with a sliding window approach for segmentation. Detailed settings are provided in Table 2.

**Table 2.** Training protocols.

| Network initialization | Random |
|---|---|
| Batch size | 6 |
| Resized size (Phase_1) | $128 \times 128 \times 128$ |
| Patch size ($Phase2_{model}$) | $96 \times 96 \times 96$ |
| Total epochs | 150 |
| Optimizer | AdamW |
| Initial learning (lr) | $3e^{-4}$ |
| Lr decay schedule | Cosine annealing |
| Training time for each model | 36 hours |
| Loss function | Dice loss and Cross entropy loss |
| model parameters ($Phase1_{model}$ / $Phase2_{model}$) | 1.36 M / 9.37 M |
| Number of flops ($Phase1_{model}$ / $Phase2_{model}$) | 1.03 G / 71.13 G |

When using the abdominal multi-organ pseudo-labels provided by the organizers, which were inferred by the FLARE22 winning algorithm, we observed that the model did not achieve the desired accuracy on the validation leaderboard. Through careful observation of these pseudo-labels, we discovered that the segmentation of certain organ categories exhibited discontinuities, indicating that the accuracy was limited by the quality of the annotations, as shown in Fig. 3.

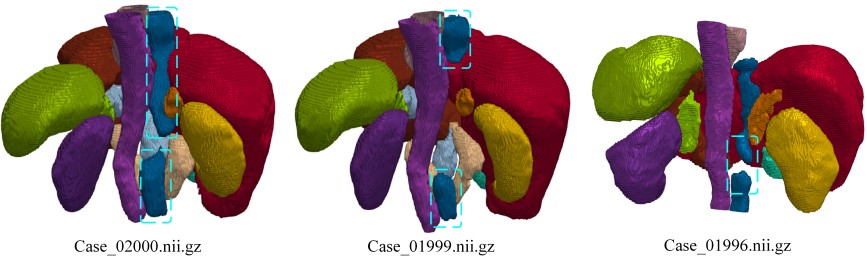

Case_02000.nii.gz      Case_01999.nii.gz      Case_01996.nii.gz

**Fig. 3.** The dashed lines indicate the instances of segmentation discontinuities.

To mitigate the adverse impact resulted by pseudo-label on segmentation performance, we applied label combine mechanism, as illustrated in Algorithm 1. Initially, we utilized the pseudo-labels $D_{initial}$ provided by the organizers to train

---

**Algorithm 1:** Pseudo-labeling iterative process

---

 **Iters**  : iterations $ItersNumbers = 4$
 **Input** : Initial labels $D_{initial}$, Initial trained model $Phase2_{model}$
 **Output:** Fine-tuned model $Phase2_{model}$

**1** **For** *iter in [1, ItersNumbers]* **do**
**2**   Getting the inference data $D_{inference}$:
**3**    $D_{inference} = Phase2_{model}(D_{initial})$ ;
**4**   $D_{newcombine} = np.zeros\_like(D_{initial})$;
**5**   **For** *class in [1, 13]* **do**
**6**    $D_{newcombine}[(D_{initial} == class)|(D_{inference} == class)] = class$;

**7**   $D_{newcombine}[(D_{initial} == 10)|(D_{inference} == 10)] = 10$;
**8**   $D_{newcombine}[(D_{initial} == 9)|(D_{inference} == 9)] = 9$;
**9**   $D_{newcombine}[(D_{initial} == 8)|(D_{inference} == 8)] = 8$;
**10**   Updating the $D_{initial}$:
**11**    $D_{initial} = D_{newcombine}$;
**12**   Fine-tuning the $Phase2_{model}$:
**13**    $Phase2_{model} = Phase2_{model}(D_{initial})$;
**14** **return** $Phase2\_model$;

---

the network of Phase2 and subsequently performed inference to obtain $D_{inference}$ from the trained model. We then combined $D_{initial}$ and $D_{inference}$ to aggregate the labels, resulting in new labels $D_{newcombine}$. Upon examining $D_{newcombine}$, we observed that the label of stomach (category 11) may replace Gallbladder and Left Adrenal Gland, both of which had inherently low accuracy on the validation submission. So, further refinement of $D_{newcombine}$ is necessary. Based on the spatial correlation between organs and the segmentation performance on the validation submission, we modified some labels in $D_{newcombine}$. Finally, we pre-trained the $Phase2\_model$ using the adjusted $D_{newcombine}$. Overall, this process was executed four times and yielded favorable results.

## 4  Results and discussion

In this section, we comprehensively analyze the proposed method from both qualitative and quantitative perspectives. For accuracy, we use the DSC and NSD metrics mentioned in Section 3.1 for evaluation. For inference efficiency, we tested the inference time for several cases. The detailed explanation of the relevant content is provided below.

### 4.1  Quantitative results

**Ablation experiment** First, to assess the impact of the two-stage algorithm on inference speed, we compared it with a model that employs only a single stage. The results are presented in Table 3. When performing abdominal multi-organ

**Table 3.** Ablation analysis of different network architectures. Evaluation CPU: 12th Gen Inter(R) Core(TM) i9-12900K CPU @ 5.2GHz × 48.

| Structure | DSC (%) | NSD (%) | Time (s) |
|---|---|---|---|
| One stage | 87.93 ±7.02 | 92.90 ±5.50 | 92.63 |
| two stage | 87.91 ±7.00 | 92.88 ±5.52 | 16.34 |

**Table 4.** Ablation evaluation of segmentation methods. Evaluation CPU: 12th Gen Inter(R) Core(TM) i9-12900K CPU @ 5.2GHz × 48.

| Methods | Spatial spacing | DSC (%) | NSD (%) | Time (s) |
|---|---|---|---|---|
| [16] | (1.5 mm, 1.5 mm, 2.0 mm) | 88.13 ±7.97 | 93.44 ±8.29 | 25.64 |
| Ours† | (2.0 mm, 2.0 mm, 2.0 mm) | 87.57 ±9.46 | 93.32 ±9.90 | 12.01 |
| Ours‡ | (1.5 mm, 1.5 mm, 2.0 mm) | 87.91 ±7.00 | 92.88 ±5.52 | 16.34 |

†and ‡represent different spatial spacing used in our method, respectively.

segmentation using the two-stage algorithm, the inference time on the CPU was significantly reduced, demonstrating the rationality of our design.

Next, to comprehensively evaluate our methods, which have spatial spacing of (1.5 mm, 1.5 mm, 2 mm) and (2 mm, 2 mm, 2 mm) respectively, in terms of both accuracy and inference speed, we compared them with the work of last year[16] on public validation sets. The results are shown in Table 4.

Although last year's work[16] achieved the highest DSC and NSD, its inference speed on the CPU is relatively slow compared to our methods. Additionally, by observing the running time, we found that some cases with a high number of slices even exceed 60 seconds. For accuracy, the methods achieved comparable performance. Therefore, we can naturally conclude that our lightweight model, along with the preprocessing steps designed by ours, effectively reduces the inference time. To further optimize inference time, we adjusted the spatial spacing to (2.0 mm, 2.0 mm, 2.0 mm) and found that this adjustment resulted in a further reduction of 4.33 seconds in running time. However, this adjustment led to the DSC decreased by 0.34%. Taking the results into comprehensive consideration, our method employs the spatial spacing of (1.5 mm, 1.5 mm, 1.5 mm) as the final selection for the preprocessing.

**Accuracy analysis** Next, to analyze the accuracy of the proposed model, we conducted experiments on public validation sets, and online validation sets. The detailed results are presented in Table 5. With regards to abdominal multi-organs segmentation, DSC and NSD achieved accuracies of (87.91, 90.02) and (92.88, 95.51), respectively. The right adrenal gland, left adrenal gland, and gallbladder exhibit high standard variance, which is attributed to their small size and the ambiguous boundaries with adjacent organs. Moreover, the small variations in accuracy across different sets further validate the model's generalization capability.

**Table 5.** Quantitative accuracy evaluation results for abdominal multi-organs.

| Target | Public Validation | | Online Validation | |
|---|---|---|---|---|
| | DSC (%) | NSD (%) | DSC (%) | NSD (%) |
| Liver | 97.40 ±1.70 | 98.23 ±4.11 | 97.66 | 98.95 |
| Right Kidney | 93.72 ±14.29 | 93.94 ±15.34 | 94.29 | 95.00 |
| Spleen | 96.82 ±2.27 | 98.17 ±5.27 | 96.51 | 98.24 |
| Pancreas | 89.61 ±3.47 | 97.88 ±2.85 | 86.75 | 96.72 |
| Aorta | 93.92 ±4.53 | 97.86 ±5.07 | 94.85 | 98.61 |
| Inferior vena cava | 88.62 ±10.45 | 90.40 ±12.50 | 90.19 | 93.11 |
| Right adrenal gland | 79.11 ±20.95 | 91.29 ±23.62 | 85.28 | 97.24 |
| Left adrenal gland | 79.23 ±23.18 | 90.34 ±26.96 | 85.50 | 97.07 |
| Gallbladder | 75.89 ±36.11 | 77.85 ±37.48 | 85.44 | 87.98 |
| Esophagus | 84.41 ±18.36 | 92.59 ±20.26 | 82.22 | 92.47 |
| Stomach | 92.19 ±14.70 | 95.05 ±16.06 | 94.59 | 97.56 |
| Duodenum | 83.69 ±9.64 | 94.13 ±7.27 | 82.93 | 93.63 |
| Left kidney | 88.19 ±22.42 | 89.65 ±20.97 | 94.06 | 95.11 |
| Average | 87.91 ±7.00 | 92.88 ±5.52 | 90.02 | 95.51 |

**Segmentation efficiency results on validation set** Since the proposed method needs to be executed on a laptop, it is necessary to consider the inference latency. Table 6 provides the inference efficiency results for some examples, all of which achieved segmentation on laptop in approximately 20 seconds. The detailed results are shown below.

**Table 6.** Quantitative evaluation of segmentation efficiency on the running time. Evaluation CPU: 12th Gen Inter (R) Core (TM) i9-12900K CPU @ 5.2GHz × 48.

| Case ID | Image Size | Running Time (s) |
|---|---|---|
| 0059 | (512, 512, 55) | 16.06 |
| 0005 | (512, 512, 124) | 19.88 |
| 0159 | (512, 512, 152) | 20.48 |
| 0176 | (512, 512, 218) | 18.11 |
| 0112 | (512, 512, 299) | 22.47 |
| 0135 | (512, 512, 316) | 23.22 |
| 0150 | (512, 512, 457) | 19.19 |
| 0134 | (512, 512, 597) | 27.03 |

**Segmentation efficiency results on test set** To ensure a fair comparison of the advantages of the algorithm in terms of robustness and inference efficiency, we further evaluated it against those proposed by other teams on the test dataset with regional variations. The results are presented in Table 7. Our algorithm achieved optimal accuracy performance across different regional populations without significantly affecting inference speed. Moreover, the algorithm

demonstrated satisfactory efficiency when performing three-dimensional medical data segmentation using only the CPU, in comparison to inference executed on the GPU.

**Table 7.** Performance comparison of different algorithms across various regional populations.

| Team Name | Asian | | | | | |
|---|---|---|---|---|---|---|
| | DSC | | NSD | | Time | |
| | Mean (%) | Median (%) | Mean (%) | Median (%) | Mean (s) | Median (s) |
| gmail | 86.2 ±6.8 | 88.6 | 92.4±6.2 | 94.9 | 32.6±6.2 | 33.7 |
| hanglokai | 87.2 ±6.6 | 90.4 | 93.1±6.1 | 95.6 | 33.3±8.9 | 34.3 |
| lyy1 | 85.2 ±6.2 | 87.8 | 92.1±5.9 | 94.3 | 15.5±3.7 | 13.7 |
| miami | 86.2 ±6.9 | 89.1 | 92±6.2 | 94.6 | 31.4±5.3 | 30.6 |
| nichtlangfackeln | 73.9 ±12.5 | 76.6 | 79.3±14.1 | 83 | 26±6.8 | 25.5 |
| fzu312chy | 61.1 ±8.8 | 62.9 | 61.6±10 | 63.6 | 35.1±12.7 | 33.4 |
| care | 73 ±11.1 | 75.2 | 78.6±12.6 | 81.1 | 268.6±64.7 | 257 |
| lyybooster | 86.6 ±6.4 | 89.3 | 92.9±5.7 | 95.1 | 24.7±2.5 | 24.4 |
| **Team Name** | **European** | | | | | |
| | DSC | | NSD | | Time | |
| | Mean (%) | Median (%) | Mean (%) | Median (%) | Mean (s) | Median (s) |
| gmail | 87.4 ±8 | 90 | 92.8±7.9 | 95.9 | 33.6±10.3 | 34.3 |
| hanglokai | 89.1 ±6 | 91.5 | 94.2±6 | 96.7 | 38.1±12.4 | 34.9 |
| lyy1 | 87.4 ±6.2 | 89.7 | 93.4±6.1 | 95.7 | 16.4±4.6 | 17.5 |
| miami | 87 ±8.4 | 89.6 | 91.8±8.6 | 95.3 | 30.6±8.1 | 29.5 |
| nichtlangfackeln | 78.2 ±13.5 | 82.1 | 82.7±14.9 | 87 | 24.2±8.4 | 24.9 |
| fzu312chy | 63.4 ±9.9 | 66.3 | 63±11.4 | 65.9 | 42,7±12.9 | 40.6 |
| care | 76.7 ±11.8 | 80.2 | 81.8±12.5 | 85 | 291.5±110.5 | 264.4 |
| lyybooster | 88.5 ±6.2 | 90.7 | 93.9±6.1 | 96.2 | 24.8±3.1 | 25.2 |
| **Team Name** | **North American** | | | | | |
| | DSC | | NSD | | Time | |
| | Mean (%) | Median (%) | Mean (%) | Median (%) | Mean (s) | Median (s) |
| gmail | 87.6 ±4.9 | 89.4 | 93±5.2 | 94.7 | 27.2±6.8 | 26.9 |
| hanglokai | 89.2 ±4.4 | 90.7 | 93.8±4.6 | 95.4 | 35.3±10.8 | 34.8 |
| lyy1 | 87.6 ±4.5 | 89.71 | 93.1±6.1 | 94.7 | 12.6±2.7 | 13 |
| miami | 87.4 ±4.4 | 88.9 | 92.5±8.6 | 94.1 | 30.1±7.8 | 29.3 |
| nichtlangfackeln | 70.7 ±15.5 | 75.8 | 873.1±17.5 | 79.2 | 20.8±9.7 | 18.1 |
| fzu312chy | 59.4 ±7.5 | 60.3 | 57.5±8.5 | 58.9 | 34.9±9.9 | 32.4 |
| care | 76.4 ±11.8 | 79.5 | 79.7±13.2 | 83.7 | 204.5±50.1 | 172 |
| lyybooster | 88.7 ±4.4 | 90.4 | 94.0±4.7 | 95.6 | 22.6±1.8 | 21.6 |

To ensure a fair comparison of the advantages of the algorithm in terms of robustness and inference efficiency, we further evaluated it against those proposed by other teams on the test dataset with regional variations. The results are presented in Table 7. Our algorithm achieved optimal accuracy performance across different regional populations without significantly affecting inference speed. Moreover, the algorithm demonstrated satisfactory efficiency when performing

three-dimensional medical data segmentation using only the CPU, in comparison to inference executed on the GPU.

Based on the analysis of the accuracy and runtime of the proposed method, we conclude that it achieves satisfactory accuracy with a relatively fast inference speed on a laptop. This indirectly indicates that the method is suitable for devices with limited computational resources.

## 4.2   Qualitative results on validation set

For a clearer observation of the segmentation performance of the proposed method, Fig. 4 provides the results of several cases on the public validation set. We observed that our methods produced segmentation results that were closely consistent with the ground truth labels in both case0036 and case0003. However, for the remaining two cases, our method encountered issues: in case0047, the liver segmentation was incomplete, and in the other, the liver was mistakenly classified as the gallbladder. The former error is attributed to the presence of large tumors within the liver, which resulted in significant voxel value deviations from the liver region. The latter error is due to the similarity between the voxel values of liver tumors and those of the gallbladder.

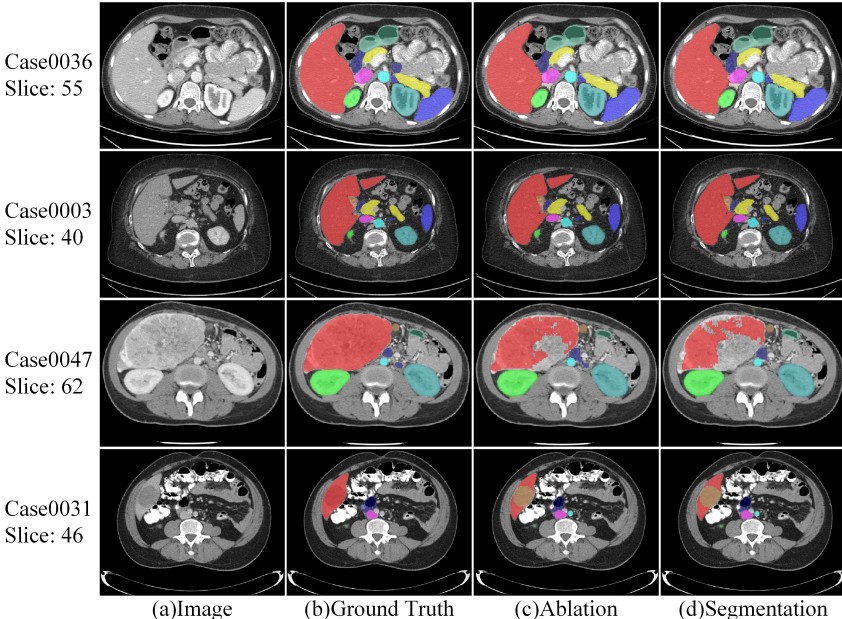

**Fig. 4.** Segmentation results on several cases from the public validation set.

### 4.3   Limitation and future work

Although our method achieves satisfactory accuracy in abdominal multi-organ segmentation tasks, it still has some certain limitations. Firstly, it struggles to achieve good segmentation for organs containing tumors. Secondly, the segmentation accuracy is limited for small target organs and those organs that are in close proximity to each other. Finally, regarding inference speed, we believe that our method could be further improved by leveraging lightweight techniques and appropriate pixel spacing settings.

## 5   Conclusion

In this paper, we propose a cascaded two-phase method to address real-time multi-organ segmentation tasks on laptop. In Phase One, we quickly localize the abdominal region. In Phase Two, we further improve last year's method by incorporating lightweight techniques. Additionally, to avoid excessive preprocessing time, we meticulously adjusted the preprocessing steps to reduce computational. The feasibility of the proposed method was validated through experiments. Through extensive observations, we identified ways to further enhance pseudo-labeling, which is crucial for segmentation accuracy. As for inference efficiency, we believe there is still room for improvement in the model, particularly through lightweight techniques and appropriate spatial pixel settings.

**Acknowledgements** The authors of this paper declare that the segmentation method they implemented for participation in the FLARE 2024 challenge has not used any pre-trained models nor additional datasets other than those provided by the organizers. The proposed solution is fully automatic without any manual intervention. We thank all data owners for making the CT scans publicly available and CodaBench [33] for hosting the challenge platform.

The study was supported by National Natural Science Foundation of China (81827805, 82130060, 61821002, 92148205), National Key Research and Development Program (2018YFA0704100, 2018YFA0704104). The project was funded by China Postdoctoral Science Foundation (2021M700772), Zhuhai Industry-University-Research Collaboration Program (ZH22017002210011PWC), Jiangsu Provincial Medical Innovation Center (CXZX202219), Collaborative Innovation Center of Radiation Medicine of Jiangsu Higher Education Institutions, and Nanjing Life Health Science and Technology Project (202205045). The funding sources had no role in the writing of the report, or decision to submit the paper for publication.

## Disclosure of Interests

The authors declare no competing interests.

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

**Table 8.** Checklist Table. Please fill out this checklist table in the answer column.

| Requirements | Answer |
|---|---|
| A meaningful title | Yes/No |
| The number of authors ($\leq$6) | Number |
| Author affiliations and ORCID | Yes/No |
| Corresponding author email is presented | Yes/No |
| Validation scores are presented in the abstract | Yes/No |
| Introduction includes at least three parts: background, related work, and motivation | Yes/No |
| A pipeline/network figure is provided | Figure number |
| Pre-processing | Page number |
| Strategies to improve model inference | Page number |
| Post-processing | Page number |
| The dataset and evaluation metric section are presented | Page number |
| Environment setting table is provided | Table number |
| Training protocol table is provided | Table number |
| Ablation study | Page number |
| Efficiency evaluation results are provided | Table number |
| Visualized segmentation example is provided | Figure number |
| Limitation and future work are presented | Yes/No |
| Reference format is consistent. | Yes/No |