# OpenReview forum: "A highly efficient segmentation method for abdominal multi-organs on laptop"
_MICCAI.org/2024/Challenge/FLARE — FLARE 2024 withMinorRevisions_

### Official Review · Reviewer_Lt7n · 2025-01-20
**Good paper**

**Rating:** 9
**Confidence:** 5

**Review:**

This paper presents a two-phase strategy for efficient abdominal multi-organ segmentation on laptops. Phase 1 quickly localizes the abdominal region using a lightweight network, while Phase 2 performs detailed segmentation using a network with Scale-Aware Modulator and Multi-head Self-Attention, featuring a novel Lightweight Attention-based Convolutional Block in the decoder. The method achieves good accuracy (DSC 90.02%, NSD 95.51%) with fast inference speed (16.34s) on laptop CPUs through optimized preprocessing and efficient network design.

Overall, this is a very complete paper. Here are some minor suggestions:
- Adding comparative experiments on whether to use pseudo-labels and single-phase vs two-phase approaches would be interesting.

---

> ### Author Response · Authors · 2025-03-25
> **Additional experiments have been conducted**
>
> Thank you for your suggestions. Below are the additional experiments I have included in the paper:
>
>     1) A comparison of the performance between the single-stage model and the two-stage model in tabel 3;
>     2)Regarding the question of whether to use pseudo-labels in the experiments, as you know, if I train the model using a validation set with ground truth, I will not be able to compare the effect of using pseudo-labels on the accuracy of the validation set, as it may lead to overfitting. This experiment could be conducted if a test set were available, but the competition organizers have not released the test set data, for which I kindly ask for your understanding.

---

### Official Review · Reviewer_SGHk · 2025-01-23
**Good Paper**

**Rating:** 9
**Confidence:** 5

**Review:**

This method achieves efficient inference speed on laptops while maintaining segmentation accuracy comparable to high-performance devices, demonstrating excellent practicality and making it highly suitable for resource-constrained real-world medical scenarios. The paper thoroughly validates the effectiveness of the method using multiple datasets and evaluation metrics, such as DSC and NSD, along with ablation experiments and runtime efficiency evaluations. Furthermore, the proposed two-phase cascaded network framework, combined with lightweight design, excels in balancing efficiency and accuracy.

---

> ### Author Response · Authors · 2025-03-25
> **Response to the reviewer’s comments**
>
> Thank you for acknowledging the work presented in the paper. Moving forward, we will conduct further research on how to execute this task more efficiently.

---

### Official Review · Reviewer_hCXP · 2025-01-27
**Review of  "A highly efficient segmentation method for abdominal multi-organs on laptop"**

**Rating:** 9
**Confidence:** 5

**Review:**

This paper presents a two-phase strategy for precise and real-time segmentation of abdominal multi-organs using a lightweight model designed for laptops. The method involves a quick localization of the abdominal region in the first phase and a detailed segmentation in the second phase, incorporating a Lightweight Attention-based Convolutional Block (LACB) to enhance inference efficiency. The study demonstrates promising performance with an average DSC of 90.02% and NSD of 95.51%, achieving an average running time of 16.34 seconds on a laptop.
There are some minor suggestions:
1. Adding more cases in Fig.3 for clear explanation.

---

> ### Author Response · Authors · 2025-03-25
> **Modifications to Figure 3**
>
> Thank you for your suggestion. I have reviewed multiple cases and found that the majority of the pseudo-labels exhibit this issue. Therefore, it can be concluded that the incompleteness of pseudo-labels limits the improvement in segmentation performance. Additionally, we have included the visualization of another case in Figure 3.

---

### Official Review · Reviewer_YcNC · 2025-03-11
**Typos and style**

**Rating:** 9
**Confidence:** 5

**Review:**

Typos and style: There are many typos or formatting issues, such as:
Complete the GitHub link;
“...using single Graphics Processing Unit(GPU)...” Missing space;
 “...Xie et al.[32] and Gao et al.[6] leverage the advantages of...” Missing space;
“of two networks(see Figure 1), as demonstrated by” Change “Figure 1” to “Fig. 1”;
“...the Dice loss function (Equation 1) and the cross-entropy loss function (Equation 2)...” The description does not correspond to the formula, and the "*" in formula 2 is changed to "×"
Please carefully read the whole paper and improve the writing quality.

---

> ### Author Response · Authors · 2025-03-25
> **Modifications to the paper format**
>
> Thank you for your suggestion. I have made revisions to the entire paper addressing the formatting issues you raised and have carefully checked for any other errors.

---

### Decision · Program_Chairs · 2025-03-20

**Decision:**

Accept

**Comment:**

Minors: Add code link to Table 1 or delete the last row.

---

> ### Author Response · Authors · 2025-03-25
> **Paper revisions**
>
> Thank you for your suggestion. I have made the corresponding revisions to the paper based on the feedback from each reviewer.
> 1)Modify the code address in the abstract
> 2）correct formatting errors in the manuscript
> 3）add more cases to Figure 3
> 4） introduce a new ablation experiment in Table 3
> 5） Add the execution results on the test set, as presented in Table 7.